# Determinants of Increased Effort of Breathing in Non-Intubated Critical COVID-19 Patients

**DOI:** 10.3390/medicina58081133

**Published:** 2022-08-21

**Authors:** Vaidas Vicka, Elija Januskeviciute, Justina Krauklyte, Aiste Aleknaviciene, Donata Ringaitiene, Ligita Jancoriene, Jurate Sipylaite

**Affiliations:** 1Clinic of Anaesthesiology and Intensive Care, Institute of Clinical Medicine, Faculty of Medicine, Vilnius University, M. K. Ciurlionio st. 21, LT-03101 Vilnius, Lithuania; 2Faculty of Medicine, Vilnius University, M. K. Ciurlionio st. 21, LT-03101 Vilnius, Lithuania; 3Clinic of Infectious Diseases and Dermatovenerology, Institute of Clinical Medicine, Faculty of Medicine, Vilnius University, M. K. Ciurlionio st. 21, LT-03101 Vilnius, Lithuania

**Keywords:** effort of breathing, silent hypoxemia, COVID-19

## Abstract

*Background and objectives:* Acute respiratory distress syndrome (ARDS) is the most common complication occurring in COVID-19 patients admitted to the ICU. Given the increased respiratory work of these patients, it is necessary to evaluate their actual breathing efforts. The aim of this study is to report the incidence and determinants of increased effort of breathing (EOB) in critical COVID-19 patients. *Materials and Methods:* This was a retrospective study of COVID-19 patients admitted to the ICU during the year of 2020. Respiratory rate (RR) was chosen as an indicator of EOB. The cut-off value was set at more than 20 breaths per minute. ROC-AUC analysis was performed to identify the accuracy of the PaO2 and PaCO2 to determine increased EOB. Furthermore, multivariate regression analysis was performed to reveal the determinants of increased EOB. *Results:* 213 patients were included in the study. Mean RR in the population was 24.20 ± 6.28. 138 (64.8%) of the patients had increased EOB. The ROC-AUC analysis revealed the PaO2 (0.656 (CI 95%: 0.579–0.734, *p* < 0.001) as more accurate predictor of EOB than PaCO2 (0.584 (CI 95%: 0.505–0.662, *p* = 0.043). In the final multivariate model, the SpO2 (exp(B) = 0.922, CI 95%: 0.874–0.97 *p* = 0.033), PaO2/FiO2 ratio (exp(B) = 0.996, CI 95%: 0.922–1.000, *p* = 0.003) and PaO2 (exp(B) = 0.989 CI 95%: 0.982–0.996 *p* = 0.003) prevailed as independent predictors of increased EOB. *Conclusions:* To conclude, PaO2 was revealed as a more accurate predictor of increased EOB than PaCO2. Further investigation revealed the independent determinants of EOB: blood oxygen saturation, PaO2 and PaO2/FiO2 ratio.

## 1. Introduction

A novel coronavirus was found to be a cause of clustered pneumonia cases in Wuhan China towards the end of 2019. The outbreak spread globally and has been considered as a pandemic by the WHO since 11 March 2020. As of 7 September 2021, a cumulative total of around 220 million confirmed cases of coronavirus disease 2019 (COVID-19) were reported with a total 4.5 million deaths worldwide [1]. Even though the majority of patients who present with COVID-19 have a mild or uncomplicated disease course, most centres report that around 10–20% will develop a severe infection requiring hospitalisation and oxygen therapy or even transfer to intensive care unit (ICU) [2]. Studies confirm that prevalence of mortality in patients with coronavirus disease in ICU is 41.6% [3].

Acute hypoxemic respiratory failure due to acute respiratory distress syndrome (ARDS) is the most common complication occurring in 60–70% of COVID-19 patients admitted to the ICU [4]. It is known that COVID-19 related ARDS (C-ARDS) has both similarities and differences compared to ARDS of other aetiologies [5]. The objects of current research are interactions between hypoxemia, increased respiratory drive and dyspnoea in COVID-19 patients. The latter is sometimes observed to be absent in the early stages of the disease (called “silent hypoxemia”) [6,7]. Considering the importance of respiratory failure in COVID-19 pathophysiology, breathing support strategies and associated potential lung damage has also been a matter of interest since the beginning of the pandemic. Invasive mechanical lung ventilation is a concern for healthcare professionals because of the increasing risk of ventilator induced injury and ventilator associated pneumonia [8]. On the other hand, there is an ongoing controversy stating that increased respiratory effort in spontaneously breathing patients may cause patient self-inflicted lung injury. Thus, the dilemma of when to switch from spontaneous breathing to invasive lung ventilation strategies and vice versa still remains.

Given the increased respiratory work of critical COVID-19 patients during the disease course, it is necessary to evaluate their actual breathing efforts. The majority of COVID-19 patients may not have tachypnoea or dyspnoea, but lung CT scans show that most of the lung tissue is damaged and does not function as it does in a healthy person [7]. The term silent hypoxemia has already been mentioned and describes this phenomenon [9,10]. The exact causes of this pathology have not yet been determined, but it is thought that one of the reasons could be that the SARS-CoV-2 virus may influence blood circulation in lungs or induce a systemic infalamtory response syndrome (SIRS) related to the dysregulation of breathing control. However, it is not known how these pathophysiological processes are related to the physiological control of breathing via partial pressure of carbon dioxide and oxygen. Therefore, the aim of this study is to report the incidence and determinants of increased effort of breathing in critical COVID-19 patients.

## 2. Materials and Methods

### 2.1. Study Sample

This was a retrospective study using a prospectively gathered institutional database. All the adult COVID-19 patients admitted to the intensive care unit during the year of 2020 were included, if they were not intubated. The sample was gathered in Vilnius University Hospital Santaros Clinics, Vilnius, Lithuania. Vilnius Regional Biomedical Research Ethics Committee permit (Reg. N. 2020/6-1233-718) was obtained for the study.

### 2.2. Effort of Breathing

Effort of breathing (EOB) was evaluated within an hour of admission to the intensive care unit. Respiratory rate was chosen as an indicator of effort of breathing, the cut-off value of increased EOB was set at more than 20 breaths per minute. Therefore, two groups (normal effort of breathing and increased effort of breathing) were formed. Respiratory rate was calculated before the start of any type of non-invasive ventilation.

### 2.3. Determinants of Effort of Breathing

Usual comorbidity data, according to the Charlson co-morbidity index definitions, was gathered. Additionally, disease length before the ICU was indicated.

Furthermore, clinical signs upon admission were assessed. The emphasis on physiological determinants of respiratory rate was made, forming four groups of clinical phenotypes of respiratory drive, using arterial blood gas analysis (Table 1). The cut-off value for high PaCO2 was used 35 mmHg, the cut-off value for low PaO2 used was 60 mmHg. Lastly, the results of the laboratory tests were gathered.

### 2.4. Statistical Analysis

#### 2.4.1. Descriptive Analysis

Statistical analysis was carried out by the SPSS statistical software package version 26.0 (IBM/SPSS, Inc., Chicago, IL). Baseline characteristics were defined using descriptive statistics. Categorical variables were stated as an absolute number (*n*) and a relative frequency (%), and continuous variables were represented as a median (interquartile range) or as a mean (± SD), depending on the normality of the distribution. The normality of distribution was tested by one sample Kolmogorov–Smirnov test.

#### 2.4.2. Comparison of Two Groups of Normal and Increased Effort of Breathing

To compare the categorical variables, Chi-square and Fisher’s exact test were performed. To compare the continuous variables, the independent samples t-test was used for the normally distributed data, and the Mann–Whitney test was used for the non-parametric data. To compare the respiratory rate in the four clinical groups of respiratory drive, an ANOVA analysis was performed.

#### 2.4.3. Accuracy Testing

Receiver operating characteristics (ROC) were measured, and area under the ROC curves (AUC) were examined to identify the accuracy of discrimination of the PaO2 and PaCO2 to determine increased effort of breathing.

#### 2.4.4. Regression Analysis

To determine the independent predictive value of all the risk factors, forward logistic regression analysis was performed.

## 3. Results

### 3.1. Study Population

Two hundred and thirteen patients were included in the study, of which 61% were male. The overall age of the patients was 61.34 years. Most of the patients were aged 50–70 years old (54.9%). The observed mortality was 42.8%. Baseline characteristics are presented in Table 2.

### 3.2. Effort of Breathing: Incidence and Descriptive Characteristics

The mean respiratory rate in the population was 24.20 ± 6.28. The distribution of respiratory rate is presented in Figure 1. In total, 138 (64.8%) patients had increased effort of breathing.

### 3.3. Effort of Breathing: Determinants

Regarding the co-morbidities and disease course, none of the variables were different across the two groups of increased respiratory effort. Among the laboratory investigations, inflammatory markers were higher in the increased EOB group: white blood cell count of 7.46 (5.10–10.49) vs. 8.68 (6.47–12.54) *p* = 0.018; CRP 106.5 (60–187) vs. 128.4 (91–215) *p* = 0.041. Furthermore, the most important differences were found in blood gas analysis and clinical signs upon admission, showing indices of oxygenation lower in the increased EOB group: SpO2 of 94 (90–97) vs 88 (80–92) *p* < 0.001; PaO2 of 81.5 (62.9–107.2) vs. 62.9 (51.8–86.4) *p* < 0.001. Other differences are detailed and presented in Table 3.

Further investigations were caried out to determine the pathophysiological changes in respiratory drive pattern using the four clinical phenotypes, described in the methods section of the article. The majority of the patients were classified in Group 4 (38.5%), while the fewest of the patients were classified in Group 1 (14.1%). The highest respiratory rate was detected in Group 3, having a statistically significant difference from the other groups with mean rate of 27.54 ± 5.0 (Table 4.)

Furthermore, the ROC-AUC analysis to determine the discriminative power of PaO2 and PaCO2 to determine the increased EOB was carried out, revealing PaO2 as the more accurate predictor, with ROC-AUC for PaO2 of 0.656 (CI 95%: 0.579–0.734, *p* < 0.001) vs. ROC-AUC for PaCO2 of 0.584 (CI 95%: 0.505–0.662, *p* = 0.043) (Figure 2).

The last step of the analysis was a multivariate regression of all the possible determinants of the increased EOB. Separate models were generated for clinical signs, arterial blood gas analysis and laboratory findings. Variables with mean values different across two different EOB groups were included in the analysis. In the clinical signs model, MAP, heart rate, SpO2, and PaO2/FiO2 ratio were included. In the final multivariate model, the SpO2 (exp(B) = 0.922, CI 95%: 0.874–0.973 *p* = 0.033) and PaO2/FiO2 ratio (exp(B) = 0.996, CI 95%: 0.922–1.000, *p* = 0.003) prevailed as independent predictors. In the blood gas model, only PaO2 remained as an independent predictor of increased EOB (exp(B) = 0.989, CI 95%: 0.982–0.996, *p* = 0.003). None of the laboratory findings were significant predictors of effort of breathing. Results are presented in Table 5.

## 4. Discussion

The main finding of this study was the importance of arterial blood oxygenation on effort of breathing. The definition of effort of breathing was based on the respiratory rate in this study. Therefore, from the physiological point of view, there should have been two main determinants: the partial pressure of oxygen in arterial blood and the partial pressure of carbon dioxide in the arterial blood, the latter being a more potent trigger in normal physiology [11]. Regarding the PaCO2, the mean value in our study was below 35 mmHg, much lower than needed to produce an acidotic shift in the tissues of the respiratory centre in the medulla, and thus to increase the respiratory rate. These results are concordant to various studies around the world, reporting no hypercarbia in COVID-19 patients [10]. However, the respiratory rate in our cohort was high, with a mean value of more than 22. This increased respiratory rate was dependent on blood oxygenation, as has been shown in the multivariate regression analysis, reporting both of the oxygenation variables (PaO2 and the SpO2) as independent predictors. These results are in line with Guyton’s physiology, which states that if the PaO2 falls below 60 mmHg, the peripheral receptors of oxygen are triggered and incite the reflex of increased breathing effort, stimulating both an increase in tidal volume and an increase in respiratory rate. To investigate these relationships more thoroughly, we decided to split the patients into four groups using the normal range values of these respiratory rate triggers, i.e., < 60 mmHg for PaO2 and > 35 mmHg for PaCO2, forming four clinical phenotypes. Interestingly, a higher respiratory rate was detected in the phenotype with low PaO2 than in the phenotype with high PaCO2. This phenomenon was again demonstrated in ROC-AUC analysis, revealing PaO2 as more accurate predictor of increased EOB. These results indicate a deranged physiology of respiration in COVID-19 patients, having hypoxemia as a more potent determinant of respiratory effort. Some explanations can be offered, for example neurotropism of the virus, which causes a neuro-inflammation and desensitisation of the respiratory centre, or systemic inflammation syndrome, which has a direct toxic effect on the respiratory centre [12]

Another determinant, revealed as a predictor of increased EOB in our patients, was PaO2/FiO2 ratio. PaO2/FiO2 was developed for and is extensively used in ARDS, having a vital place in the Berlin classification criteria [13,14]. This ratio is a direct measurement of gas exchange across the lungs and is highly indicative of the damage in the lung tissue. With regards to effort of breathing, PaO2 is used in the formula, providing the effect on the respiratory rate, which has already been discussed in this article. However, rather than evaluating the PaO2/FiO2 ratio, it should be noted that this index is associated with SIRS, which is the level of damage to the lungs reported in radiology investigations and various lung stress markers [15]. Therefore, it should not only be regarded as an index of gas exchange, but also as a marker of the damage extent. Thus, when lungs are damaged and have a severely lowered compliance, the same tidal volume that begins to trigger the stretch receptors in the lungs is activated (the Hering–Breuer reflex), and the respiratory rate is increased.

Systemic inflammation syndrome plays a vital role in the control of respiration in critical COVID-19 patients [16]. This was in part demonstrated in our study. When comparing two groups of increased and normal respiratory effort, we have found out that white blood cells count and levels of C-reactive protein were higher in the increased EOB group. These variables were not significant in further regression analysis; however, the variation of the actual values was too high and rendered these variables not suitable for regression analysis. Thus, we may stipulate that having higher levels of various inflammation markers is indicative of an ongoing deranged respiration process, despite that it is the state of the conventional regulatory system. This has been shown in other studies, evaluating the level of various biomarkers of SIRS and respiratory rate in COVID-19 patients [17]. On the other hand, the consumption of oxygen in tissues is much higher during the inflammation process, suggesting a higher demand of oxygen, a high oxygen extraction ratio, associated with lower oxygenation indices. These effects would not be as readily seen in PaCO2, since solubility of this gas is much higher, and it is easily removed even through the damaged tissue in the lungs. To sum up, the literature suggests that SIRS is associated with respiratory drive in other ways than simply through the partial pressures of PaO2 and PaCO2, which is only in part shown in our study with no definitive conclusion.

There are some limitations to our study. The main limitation is the definition of the effort of breathing. Respiratory rate is only one of contributors to the stress the lungs are exposed to. Indeed, to assume that respiratory rate is equal to work of breathing would be an oversimplification. However, there is no golden standard or method to evaluate effort of breathing in spontaneously breathing patients. Physical work, i.e., mechanical power expressed in joules per minute, can only be estimated from volume/pressure loops for mechanically ventilated patients by using the simplified and full formulas proposed by J.J. Marini. In these calculations, the energy generated per one inspiration is determined and multiplied by the respiratory rate/ventilator rate. Since it is not possible to obtain the pressure/volume curve for spontaneously breathing patients, we decided to stress the importance of the respiratory rate, in this case by simplifying it to make it more practical and usable. This is the reason why we have used the term “effort of breathing”, and avoided using the term “work of breathing”. Furthermore, we determined that we should accurately measure the tidal volume of a patient with a closed breathing circuit, because this would fumigate the patient and create potential for false measurements. Another limitation of our study is with the sample size, which resulted in most of the variables having a non-parametric distribution that we had to account for during the statistical analysis, therefore, we lost a lot of important answers. Thus, the results of our study should be regarded to as associations rather than as causations.

## 5. Conclusions

This was a retrospective study of non-intubated, spontaneously breathing, critical COVID-19 patients, focusing on their effort of breathing. With the respiratory rate cut-off value of 20, the majority of the patients had an increased effort of breathing. The independent factors of increased effort of breathing were blood oxygen saturation, PaO2, and PaO2/FiO2 ratio. Further investigation revealed PaO2 as a more accurate predictor of increased effort of breathing than PaCO2.

## Figures and Tables

**Figure 1 medicina-58-01133-f001:**
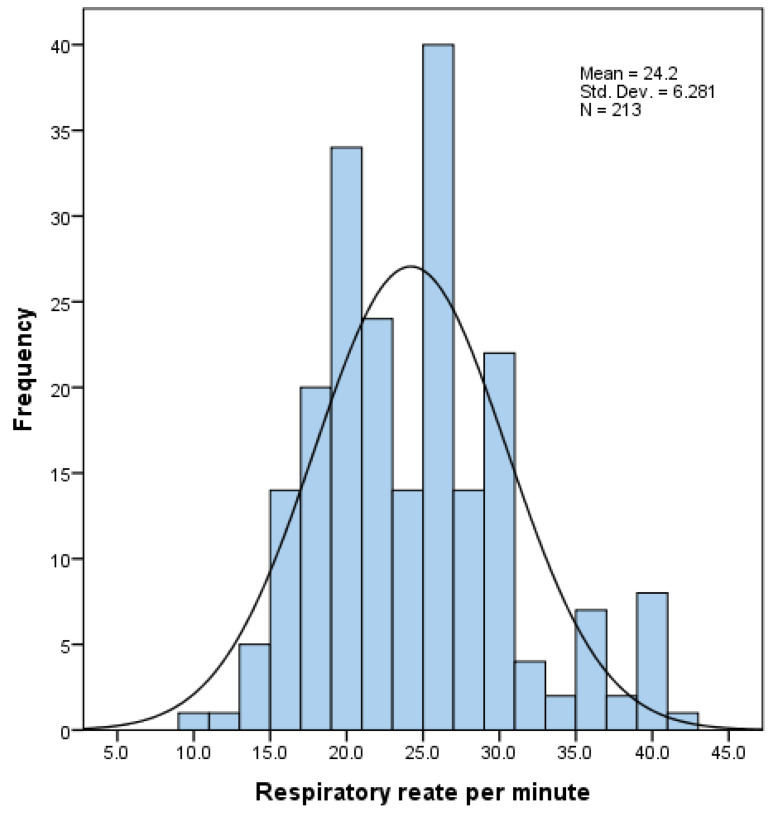
Respiratory rate distribution. The y-axis denotes the number of patients, x-axis denotes the respiratory rate.

**Figure 2 medicina-58-01133-f002:**
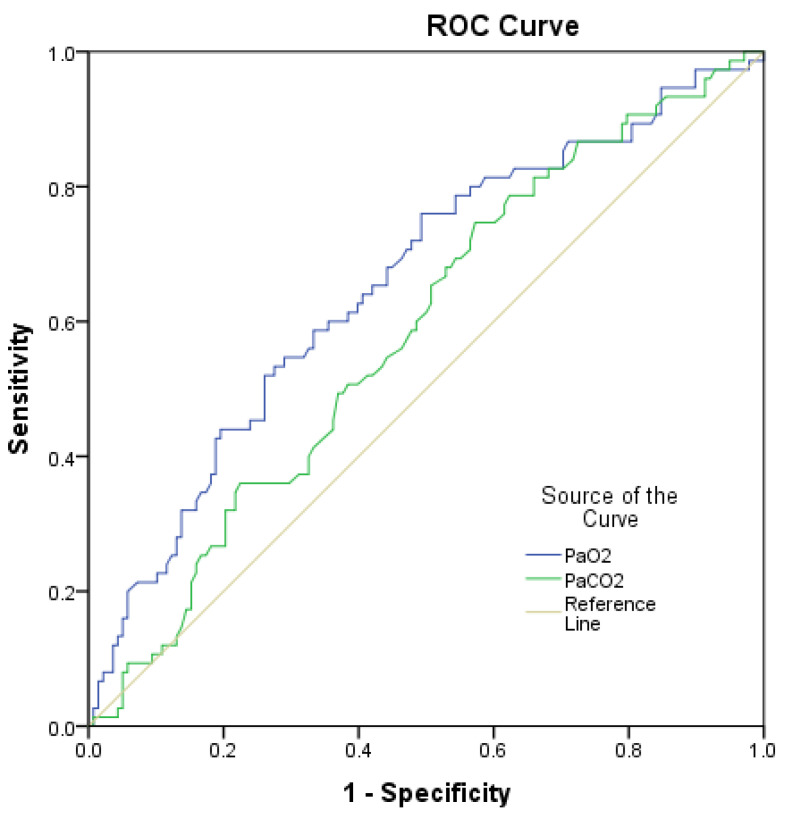
Accuracy of increased effort of breathing prediction. ROC-AUC for PaO2 of 0.656 (CI 95%: 0.579–0.734, *p* < 0.001), ROC-AUC for PaCO2 of 0.584 (CI 95%: 0.505–0.662, *p* = 0.043). PaCO2—partial pressure of carbon dioxide in arterial blood, PaO2—partial pressure of oxygen in arterial blood.

**Table 1 medicina-58-01133-t001:** Clinical phenotypes of respiratory drive.

Group 1	High PaCO2	Low PaO2
Group 2	High PaCO2	Normal PaO2
Group 3	Normal PaCO2	Low PaO2
Group 4	Normal PaCO2	Normal PaO2

Table 1 legends: PaCO2—partial pressure of carbon dioxide in arterial blood, PaO2—partial pressure of oxygen in arterial blood.

**Table 2 medicina-58-01133-t002:** Baseline characteristics of the patients.

Variable	*n* = 213, (%), Mean ± SD	Median (IQR)
Demographics		
Gender:		
Female	83 (39.0)	
Male	130 (61.0)	
Age (y)	61.34 ± 13.26	
Age groups (% within group)		
<50 y	36 (16.9)	
50–59 y	64 (30.0)	
60–69 y	53 (24.9)	
70–79 y	41 (19.3)	
>80 y	19 (8.9)	
Co-morbidities		
Obesity	68 (31.9)	
Hypertension	162 (76.1)	
Chronic cardiac disease	98 (46.0)	
CKD	72 (33.8)	
Immunosuppression	28 (13.0)	
Diabetes	67 (31.5)	
COPD	21 (9.9)	
Asthma	14 (6.6)	
Mortality risk scores		
APACHE II		12 (8–16)
ISARIC-4C		10 (7–13)
Clinical course		
MV	93 (43.7)	
Length of stay		18 (13–28)
Mortality	89 (42.8)	

CKD—Chronic kidney disease; COPD—chronic obstructive pulmonary disease; MAP—mean arterial pressure; MV—mechanical ventilation; ICU—intensive care unit; APACHE II—acute physiology and chronic health evaluation; ISARIC-4C—ISARIC-4C (International Severe Acute Respiratory Infection Consortium Clinical Characterisation Protocol) score; IQR—interquartile range; SD—standard deviation.

**Table 3 medicina-58-01133-t003:** Determinants of increased effort of breathing.

Variable	Normal Effort of Breathing *n* = 75 (35.2%)	Increased Effort of Breathing *n* = 138 (64.8%)	*p* Value
Demographics			
Gender:			0.078
Female	23 (30.7)	60 (72.3)	
Male	52 (69.3)	78 (56.5)	
Age	62.43 ± 14.13	60.75 ± 12.79	0.395
Co-morbidities			
Obesity	23 (30.7)	45 (32.6)	0.878
Hypertension	59 (78.7)	103 (74.6)	0.615
Chronic cardiac disease	37 (49.3)	61 (44.2)	0.476
CKD	27 (36.0)	45 (32.6)	0.651
Immunosuppression	7 (9.3)	21 (15.2)	0.290
Diabetes	26 (34.7)	41 (29.7)	0.537
COPD	9 (12.0)	12 (8.7)	0.475
Asthma	4 (5.3)	10 (7.2)	0.774
	**Median (IQR)**	**Median (IQR)**	
Disease course			
Time to ICU	8.5 (4.75–12)	8 (4–14.5)	0.183
Clinical signs upon admission			
Fever	36.6 (36.6–37.2)	36.9 (36.6–37.5)	0.219
MAP	88.0 (80.0–98.3)	96.7 (84.8–105.)	0.010
Heart rate	80 (70–92)	87.5 (76–101)	0.011
SpO2	94 (90–97)	88 (80–92)	<0.001
PaO2/FiO2	186.9 (120.0–250.0)	95.9 (65.5–152.8)	<0.001
Arterial blood gas analysis			
PaO2	81.5 (62.9–107.2)	62.9 (51.8–86.4)	<0.001
PaCO2	35.1 (31.9–38.9)	32.8 (28.9–36.6)	0.043
HCO3	22.3 (20.6–25.3)	22.4 (19.1–24.5)	0.297
pH	7.44 (7.40–7.49)	7.45 (7.40–7.48)	0.361
Laboratory findings			
Urea	7.5 (5.5–10.2)	8.0 (6.2–11.6)	0.269
Creatinine	78 (61.7–106)	85 (62.5–103.5)	0.793
PCT	0.15 (0.08–0.54)	0.25 (0.13–0.81)	0.171
CRP	106.5 (60–187)	128.4 (91–215)	0.041
D-dimer	720 (410–1170)	805 (430–1647)	0.585
IL-6	27.2 (11.5–79.8)	40.7 (16.6–89.2)	0.158
Feritin	768 (425–2299)	1131 (581–2114)	0.180
Lymph. count	0.7 (0.6–0.95)	0.65 (0.5–0.9)	0.171
WBC	7.46 (5.10–10.49)	8.68 (6.47–12.54)	0.018

CKD—chronic kidney disease; COPD—chronic obstructive pulmonary disease; ICU—intensive care unit; MAP—mean arterial pressure; SpO2—blood oxygen saturation; PaO2/FiO2—ratio of partial pressure of oxygen in blood to inspired fraction of oxygen; PaO2—partial pressure of oxygen in blood; PaCO2 partial pressure of carbon dioxide in blood; HCO3—bicarbonate concentration in blood; PCT—procalcitonin; CRP—C-reactive protein; WBC—white blood cell count.

**Table 4 medicina-58-01133-t004:** Clinical phenotypes of respiratory drive.

	N (%)	Mean ± SD	CI 95%	*p* Value
Group 1	2 (0.9)	25.7 ± 8.0	22.7–28.7	<0.001
Group 2	12 (5.6)	22.2 ± 6.0	20.7–23.8	
Group 3	71 (33.3)	27.5 ± 5.0	26.0–29.1	
Group 4	128 (60.1)	24.20 ± 6.3	22.0–24.5	

Group 1: high PaCO2 and low PaO2, Group 2: high PaCO2 and normal PaO2; Group 3: normal PaCO2 and Low PaO2; Group 4: normal PaCO2 and normal PaO2. *p* value denotes ANOVA test result of the differences across the groups.

**Table 5 medicina-58-01133-t005:** Multivariate regression analysis of increased effort of breathing determinants.

	Univariate Regression			Multivariate Regression		
Variable	Exp(B)	95% CI	*p* Value	Exp(B)	95% CI	*p* Value
Clinical signs						
MAP	1.024	1.005–1.044	0.015	n.s.	n.s.	0.085
Heart rate	n.s.	n.s.	0.062	n.i.	n.i.	n.i.
SpO2	0.894	0.850–0.940	<0.001	0.922	0.874–0.973	0.033
PaO2/FiO2	0.992	0.989–0.996	<0.001	0.996	0.922–1.000	0.003
Arterial blood gas						
PaO2	0.989	0.982–0.996	0.003	0.989	0.982–0.996	0.003
PaCO2	n.s.	n.s.	0.218	n.i.	n.i.	n.i.
Laboratory						
CRP	n.s.	n.s.	0.341	n.i.	n.i.	n.i.
WBC	n.s.	n.s.	0.132	n.i.	n.i.	n.i.

Exp(B)—exponentiation of regression coefficient B; CI—confidence interval; MAP—mean arterial pressure; SpO2—blood oxygen saturation; PaO2/FiO2—ratio of partial pressure of oxygen in arterial blood to inspired fraction of oxygen; PaO2—partial pressure of oxygen in arterial blood; PaCO2 partial pressure of carbon dioxide in arterial blood; CRP—C-reactive protein; WBC—white blood cell count; n.i.—not included; n.s.—not significant.

## Data Availability

The dataset used during the current study is available from the corresponding author on reasonable request.

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
