# Peer review of "Determinants of Increased Effort of Breathing in Non-Intubated Critical COVID-19 Patients"

_medicina, 2022, doi:10.3390/medicina58081133_

Round 1

Reviewer 1 Report

Thanks for allowing me to review this manuscript.

In their work Vicka et al. correlates the “effort of breathing” (e.g. respiratory rate) to other physiological variables that are readily available, such as PaO2, PaCO2, and PaO2/FiO2.
They enrolled 213 patients and divided them according to Low/High PaO2 (cut off 60mmHg) and Low/High PaCO2 (cut off 45mmHg) to identify the sub-group with the higher respiratory rate.
Finally, the authors perform a multivariate regression to find the variables more strongly associated with respiratory rate.

However, the work has serious flaws that need to be addressed.

1.       The definition of “Effort of breathing”. In the methods the authors assume that respiratory rate = work of breathing, and somewhat arbitrarily decide that a respiratory rate > 20 means increased “effort of breathing”.

However, this is a bit of an overstretch. In the computation of work of breathing respiratory rate is often less important than inspiratory effort, which is the first one to vary in case of increased respiratory drive.
This is well elucidated by two recent reviews (Vaporidi, AJRCCM, 2020 – Spinelli, ICM, 2020): the respiratory rate increases significantly only for high respiratory drive, and only after there is an increase in tidal volume (due to an increased inspiratory effort). This is physiologically sound, has increasing minute ventilation through respiratory rate demands higher muscular effort than increasing transpulmonary effort, unless the respiratory system compliance is very low.
From this prospective, the assumption “respiratory rate = effort of breathing” is debatable and should be reconsidered in the paper.

2.       In hypoxemic patients the cut-off for “High PaCO2” should be reinterpreted, as the lung damage shifts the brain PaCO2-Ventilation curve (Vaporidi, AJRCCM, 2020 – Spinelli, ICM, 2020). In this scenario the feeling of dyspnea appears when the patients cannot match the required PaCO2 despite an intense inspiratory effort and tachypnea.

3.       The inclusion of COPD and asthmatic patients should be carefully reconsidered. These patients have different respiratory mechanics, and their respiratory drive is modulated differently from purely hypoxemic patients.

4.       For the above mentioned reason, the cut off for “high/low PaCO2” could be 35mmHg, as lower PaCO2 can imply increased inspiratory effort/tachipnea and increased respiratory drive (Grieco, AJRCCM, 2021).

5.       There is no mention to the kind of respiratory support used in this context. CPAP, NPPV and HFNO have variable effects on PaO2/FiO2, PaCO2, respiratory rate and respiratory rate, compared to standard oxygen therapy. This should be clarified.

1.       All the relationship between PaO2/FiO2 and respiratory rate/respiratory mechanics should be taken with caution. It is highly probable that patients with lower P/F are simply the most severe, therefore having higher respiratory rate, higher tidal volumes and therefore lower PaCO2. This is not a novelty in literature.

Reviewer 2 Report

Introduction

1. Please add the full form of the "SIRS" (Systemic inflammatory response syndrome) in line 65.

Materials and Methods

1. Where is the place that the study used data?

I think the sample size of 213 should come from one or two sites. Please specify the site (hospital/institute/medical centre, city, country) where you used the data.

2. Line 79. "20 inspirations per minute". That is the "respiration per minute"?

3. Please use "mmHg" instead of "mmHG" in line 87. Because Hg is the symbol of Mercury (hydrargyrum), not HG.

4. Statistical analysis. This manuscript used SPSS version 26; this programme can compute Fisher's exact test.
If possible, I suggest using Fisher's exact test preferred because an exact p-value is more suitable than an approximation p-value from the Chi-square.

5. Line 104. "Regression analysis", the first character was not italic; please revise it.

Results

1. Did you collect the history of smoke (Continue, Quit or Never) in the participants?

If you have it, suggest adding it to the demographic. 

All manuscript

1. It would be great if you could use the subscript to these words; PaCO2 (PaCOâ‚‚), PaO2 (PaOâ‚‚), FiO2 (FiOâ‚‚), HCO3 (HCO₃).

Reviewer 3 Report

This retrospective study examined non-intubated, self-breathing patients with COVID-19, particularly their respiratory effort. The paper is well conceived. No comments overall, but while I am not qualified to judge the English language and style, it could be appropriate a qualified exam of English on Manuscript.
